# Comorbid anxiety and depression: Prevalence and associated factors among pregnant women in Arba Minch zuria district, Gamo zone, southern Ethiopia

**Agegnehu Bante**[1]*, **Abera Mersha**[1], **Zerihun Zerdo**[2], **Biresaw Wassihun**[3], **Tomas Yeheyis**[4]

**1** School of Nursing, College of Medicine and Health Sciences, Arba Minch University, Arba Minch, Ethiopia, **2** Department of Medical Laboratory Science, College of Medicine and Health Sciences, Arba Minch University, Arba Minch, Ethiopia, **3** Department of Midwifery, College of Medicine and Health Sciences, Arba Minch University, Arba Minch, Ethiopia, **4** School of Nursing, College of Medicine and Health Sciences, Hawassa University, Hawassa, Ethiopia

* agegnehubante@gmail.com

**Data Availability Statement:** All relevant data are within the paper and its Supporting Information files.

## Abstract

### Introduction

Prenatal anxiety and depression are major health problems all over the world. The negative sequela of prenatal comorbid anxiety and depression (CAD) has been suggested to be higher than that of anxiety or depression alone. CAD increases the odds of preterm birth, low birth weight, prolonged labor, operative deliveries, postpartum psychiatric disorders and long term cognitive impairment for the newborn. Despite its significant ill consequences, there is a dearth of studies in low-and middle-income countries. So far, to the best of our knowledge, no study assessed the prevalence of CAD in Ethiopia. Hence, the purpose of this study was to assess CAD and associated factors among pregnant women in Arba Minch Zuria district, Gamo zone, southern Ethiopia.

### Methods

A community-based cross-sectional study was conducted among 676 pregnant women from January 01 to November 30, 2019. Patient Health Questionnaire 9-item (PHQ-9) and Generalized Anxiety Disorder 7-item (GAD-7) scales were used to assess depression and anxiety respectively. The data were collected electronically using an open data kit (ODK) collect android application and analyzed using Stata version 15.0. Bivariate and multivariable analyses were carried out to identify factors associated with CAD using binary logistic regression. Statistical significance was set at p-value < 0.05.

### Results

A total of 667 women were involved. The prevalence of CAD was 10.04% [95% confidence interval (CI): 7.76, 12.33]. Being married [adjusted odds ratio (AOR): 0.16, 95% CI: 0.05, 0.56], categorized in the highest wealth quintile [AOR: 2.83, 95% CI: 1.17, 6.84], having

**Funding:** AB received award from Arba Minch University (www.amu.edu.et), with an award number of GOV/AMU/TH/CMHS/NUR/DSS/03/10. The funders had no role in study design, data collection and analysis, decision to publish, or preparation of the manuscript.

**Competing interests:** The authors have declared that no competing interests exist.

medical illness [AOR: 3.56, 95% CI: 1.68, 7.54], encountering pregnancy danger signs [AOR: 2.66, 95% CI: 1.06, 6.67], experiencing life-threatening events [AOR: 2.11, 95% CI: 1.15, 3.92] and household food insecurity [AOR: 3.51, 95% CI: 1.85, 6.64] were significantly associated with CAD.

## Conclusions

In general, one in every ten women faced CAD in the study area. Nutritional interventions, early identification and treatment of pregnancy-related illness and medical ailments, prenatal mental health problems screening and interventions are imperative to minimize the risk of CAD in pregnant women.

## Introduction

Globally, ~10% of pregnant women suffer from a mental disorder, primarily depression; this figure is higher (16%) in underdeveloped nations [1]. Depression and anxiety are common mental disorders in pregnant women and commonly co-occurred [2]. Findings from a systematic review comprising twenty-three studies suggested that common mental disorders (CMDs) during pregnancy are ranged from 1–37%; antenatal depression and anxiety alone are ranged from 1–30% and 1–26% respectively [3]. A meta-analysis conducted by Falah and coworkers after including 66 studies from 30 countries (i.e. most of the studies were from high-income countries) reported that comorbid anxiety and depression (CAD) is prevalent and seeks medical attention; 9.5% of pregnant women had self-reported co-morbid anxiety and mild to severe depressive symptoms [4]. Thiagayson and associates also reported that 5% of Singaporean women had prenatal CAD [5]. Likewise, studies from Spain, Turkey and Pakistan also demonstrated that 9.5%, 47.6% and 13.5% of pregnant women encountered CAD respectively [6, 7].

Though the co-morbidity of anxiety and depression was not addressed, a systematic review in Africa showed that the mean prevalence of antenatal depression and anxiety were 11.3% and 14.8% respectively [8]. A study from urban South Africa demonstrated that 7.6% and 9.7% of pregnant women experience CAD in early and late pregnancy respectively [9]. In Ethiopia, neither the status of CAD nor anxiety is reported but a systematic review carried out by Getinet and colleagues reported that the pooled estimate of antenatal depression is 23.6% [10].

Previously published evidences consistently reported that low socioeconomic status, adverse events in life, poor social support, intimate partner violence (IPV), previous episode of CMDs, history of chronic medical illness, unmarried status and unplanned pregnancy increases the odds of perinatal mental disorders [2, 3, 11, 12]. Also, Premji and associates reported that a high level of perceived stress, having ≥ 3 children and adverse childhood experience increases the odds of CAD [6]. Household food insecurity also increases the odds of mental health problems during pregnancy [13–15]. Moreover, poor emotional support is reported as one of the single most predictors for CAD symptoms in a multinational study conducted in Turkey and Spain [7].

Undiagnosed and untreated CAD during pregnancy increases the odds of perinatal adverse outcomes [16]. It has been suggested that CAD has a significant negative effect on neonates than those of anxiety or depression alone [17]. CAD in the third trimester is mainly associated with oligohydramnios, intrauterine growth retardation (IUGR), diminished placental perfusion and preterm labor [18]. CAD also increases the likelihood of low birth weight (LBW), preterm birth, prolonged labor and delayed initiation of breastfeeding [19, 20]. Moreover, it

results in poor infant cognitive development [21], mental health problems in late childhood [22], birth asphyxia and coronary heart disease [23]. Furthermore, prenatal mental disorders increase the number of non-scheduled antenatal care (ANC) visits, emergency health care visits [24], maternal postpartum mental health problems and infanticide [1, 25].

Though anxiety and depression are treatable, about one-third and less than half of pregnant women received appropriate treatment respectively [26]. Especially, in low and middle-income countries (including Ethiopia) maternal psychological interventions are very limited [27]. In Ethiopia, antenatal depression is ranged from 11.8% to 31.2% [28–37]. However, little emphasis has been given to prenatal anxiety and other mental health problems including CAD. So far, to the best of our knowledge, this is the first study that investigates the comorbidity of antenatal anxiety and depression in Ethiopia. Therefore, the purpose of the study was to assess the prevalence of CAD and associated factors among pregnant women in Arba Minch Zuria district, Gamo zone, southern Ethiopia.

## Methods

### Study setting, design and period

A community-based cross-sectional study was conducted in Arba Minch Zuria district, Gamo zone, southern Ethiopia from January 01 to November 30, 2019. The district is bordered by Dirashe special woreda on the south, Bonke on the west, Dita and Chencha on the north and Oromia region on the east. Based on the 2007 census projection, the district had a total population of 164,529 in 2019; of whom 82,330 were women. Arba Minch Zuria district had a total of 31 kebeles (smallest administrative units) and it is included under Arba Minch Zuria Demographic and Health Development Program (AM-DHDP). The center was established in 2009 and became a member of INDEPTH in 2015. AM-DHDP is owned by Arba Minch University and it is one of the six public universities with Demographic and Surveillance System (HDSS) in Ethiopia [38].

### Study population

All pregnant women who were living in Arba Minch Health and Demographic Surveillance Site (AM-HDSS) were the study population. Women were enrolled in the study irrespective of their age and pregnancy trimester. Those pregnant women who were unable to communicate due to serious illness and not available after three consecutive home visits were excluded from the study.

### Sample size determination

The minimum sample size required for this study was determined by EpiInfo-7 StatCalc using single population proportion formula after fulfilling the following assumptions: 95% level of confidence, 5% margin of error, 26.2% proportion of mental distress in pregnant women [29]. After incorporating a design effect of 2 and a 10% non-response rate the final sample size for this study was 654. However, due to the nature of cluster sampling, all 676 pregnant women living in the selected clusters were included.

**Sampling procedure.** Out of 31 kebeles in Arba Minch Zuria district 9 (one small town and 8 rural Kebeles) were selected randomly and included under AM-DHDP [38]. For our study, the kebeles were considered as a cluster and all pregnant women who live in these clusters were included. Hence, the first step was identifying all pregnant women in the selected clusters. Thus, AM-HDSS and health extension worker's pregnancy register was used to identify pregnant women. Besides, to minimize the chance of missing early pregnancies; the WHO

and CDC pregnancy screening checklist was used [39]. During the study period a total of 676 pregnant women were identified from the selected clusters: 61, 41, 38, 109, 131, 66, 59, 104 and 67 from Chano Chalba, Kola Shara, Genta Meyche, Zigiti Merche, Gatse, Laka, Shelle Mella, Kolla Shelle, and Zeyse Dembele respectively.

## Study variables and measurements

The data collection instrument was organized after reviewing previous literature [5, 12, 17, 24, 40]. The tool comprises socio-demographic information (age, marital status, religion, residence, educational status, occupation and husband education and occupation), socioeconomic status (assessed using locally available assets), obstetric characteristics (gravidity, parity, age at marriage, pregnancy status, ANC visit, gestational age, pregnancy danger signs), psychological problems and psychosocial relationships (social support, interpersonal relationship and experience of life-threatening events), medical illness (hypertension, diabetes, asthma, cardiac and renal disease), household food security status, substance abuse, anxiety and depression. The tool was prepared in XLSForm in excel and converted to XForms using XLSForm online converter to collect the data using Open Data Kit (ODK) tools. Then it was uploaded to the ODK aggregate server installed in AM-HDSS. ODK aggregate is a Java application that store, analyze and present XForm survey data collected using ODK Collect, which is an android application that replaces paper forms used for data collection [41].

The Patient Health Questionnaire (PHQ-9), originally designed by Spitzer and colleagues was used to measure antenatal depression. PHQ-9 score ranges from 0 to 27, each of the 9 items can be scored from 0 (not at all) to 3 (nearly every day) [42]. PHQ-9 was validated in the Ethiopian context and it has good internal consistency (Cronbach's alpha ($\alpha$) = 0.81) and excellent intra-class correlation of 0.92 [43]. In our study, the reliability ($\alpha$) coefficient of PHQ-9 was 0.87. In the current study, a score of $\geq$ 5 was used as a cut-off point for possible antenatal depression [24].

Anxiety was assessed using the Generalized Anxiety Disorder 7-item (GAD-7) scale; which was developed by Robert L. Spitzer (MD, professor of psychiatry) [44]. Although the GAD-7 scale was not validated in the Ethiopian context, the Spanish-language version was tested as a reliable ($\alpha$ = 0.89) screening tool for antenatal anxiety [45]. The reliability coefficient of the GAD-7 scale in the current study was 0.83. GAD-7 score ranges from 0 to 21, each of the items can be scored from 0 (not at all) to 3 (nearly every day). In our study; a score of $\geq$ 5 was taken as the cut-off point to categorize pregnant women as having anxiety [45]. Participants who were screened positive both for anxiety and depression were categorized as having CAD [4].

Substance abuse was measured using the Fast Alcohol Screening Test (FAST) scale developed from the Alcohol Use Disorders Identification Test (AUDIT) by Hodgson et al., 2002. FAST has 4 items with a score ranges from 0 to 16, where a score of $\geq$3 indicates harmful drinking [46].

IPV was measured using 5 items Women's Abuse Screening Test (WAST) scale; which is originally developed by Brown et al., 1996 [47]. WAST score ranges from 0–16 where a score > 1 indicates the presence of violence [48]. Social support was measured using the Oslo Social Support Scale (OSSS-3). OSSS-3, originally developed in Neverland's, has three items and its score ranges from 3–14. It was categorized into three as "poor social support" 3–8, "moderate support" 9–11 and "strong support" 12–14 [49]. The List of Threatening Experiences (LTE) developed by Brown & Harris (1978) was used to measure life events. LTE is a self-reported questionnaire with a list of 12 threatening events that invite a dichotomous response (yes/no). The number of life events were counted for each respondent and recorded as a value from 0 to 12 [50].

Household food security was assessed using the Household Food Insecurity Access Scale (HFIAS) developed by the Food and Nutrition Technical Assistance (FANTA) Program of the U.S. Agency for International Development. It has 9 items with a follow-up question about the frequency of occurrence if the response is "yes" for the main questions. HFIAS score ranges from 0 to 27. In the current study, household food security status was categorized into two as "food secure" when the participants didn't experience food access conditions in the past 4 weeks and "food insecure" when they were unable to access sufficient food at all time to lead active and healthy life [51].

## Data collectors and data collection procedure

Twelve experienced individuals working in AM-HDSS were recruited for data collection. Besides, four MSc holder supervisors were recruited. Theoretical and practical training was given for the data collectors and supervisors for two consecutive days. ODK collect version 1.17.2 application was installed on the data collector's tablet given from the center and the blank form was downloaded from the server. Then, the tool was pre-tested on 34 pregnant women in Arba Minch town two weeks before the actual data assortment. Supervisors were kept in touch with the data collectors to regularly check the data collection procedure. Finally, the data collectors sent the completed data forms weekly to the server through a wireless connection.

## Data quality control

A structured interviewer-administered questionnaire was prepared in English and translated into Amharic and vice versa to ensure its consistency. Experienced ODK Experts were consulted at each step of the data collection. The training was given to data collectors and supervisors about how to collect data using ODK and briefed on each question included in the tool. Moreover, a pretest was done to ensure the clarity of the tool. Supervisors were checked on the spot and review all the questionnaires to ensure completeness of the forms. Furthermore, the investigators were kept in touch with the server workers to regularly check the sent files from each data collector.

## Data management and analysis

Each data files sent from the data collector's tablet were downloaded from the server and saved as an excel file. Finally, the data set were imported to Stata version 15.0 for cleaning, coding and analysis. Descriptive statistical analyses such as simple frequencies, mean and standard deviation were used to describe the characteristics of participants. The wealth index was constructed via Principal Component Analysis (PCA) and the family wealth was grouped into three quartiles. Bivariate analysis was used to see the association between each independent variable and CAD using binary logistic regression. To control confounding factors; variables with a p-value of $\leq 0.25$ in the bivariate analysis, significant in previous studies and biological plausibility were taken for the multivariable analysis. Standard error and Hosmer-Lemeshow tests were used to check multicollinearity and the model goodness of fit respectively. Adjusted odds ratio (aOR) with 95% confidence interval (CI) was used to identify factors associated with CAD. The level of statistical significance was set at a p-value < 0.05.

## Ethical considerations

Ethical approval was received from Arba Minch University, college of medicine and health sciences, Institutional Review Board (IRB) with an ethical clearance number of CMHS/

12031653/111. Moreover, this study was conducted under the Declaration of Helsinki, and the participants were informed of the purpose of the study. Furthermore, before the commencement of the data collection written informed consent was obtained from all participants. To maintain the confidentiality of information, code numbers were used throughout the study. Pregnant women who screened positive for depression or anxiety received onsite mindfulness interventions. Also, they were linked with the health extension workers for regular psychological interventions and referral to health facilities in case of severe cases like suicidal ideation.

## Results

### Socio-demographic and socioeconomic characteristics

Out of 676 pregnant women involved in the current study, 9 were excluded due to incomplete information. Hence, 667 (98.7%) pregnant women were eligible for analysis. The mean (±SD) age of the participants was 27.4 (±6) years. Of the study participants, 197 (29.6%) were within the age group of 25–29 years, 360 (54%) had no formal education, 646 (96.9%) were married, 533 (79.9%) were housewives and 650 (97.4%) were rural dwellers (**Table 1**).

### Obstetrics characteristics

Of the participants, 315 (47.2%) got pregnant for the first time at the age of 20–24 years, 546 (81.8%) were multigravida and the current pregnancy was planned for 622 (93.3%). Among the participants, 380 (57.0%) initiated ANC and 43 (6.4%) experienced pregnancy danger signs; of them, 28 (65.1%) encountered blurred vision (**Fig 1**). Out of the total pregnant women enrolled in this study; 243 (36.4%) reported IPV, 176 (26.4%) experienced LTE, 51 (7.7%) had hazardous alcohol drinking habit and 186 (27.9%) were household food insecure (**Table 2**). Moreover, 543 (81.4%) of the participants had poor social support.

### Prevalence of comorbid anxiety and depression

Overall, 10.04% [95% CI: 7.76, 12.33%] of the participants had CAD. Of the participants, 74 (11.09%) encountered depression alone (**Fig 2**).

### Factors associated with comorbid anxiety and depression

After adjustment six variables: marital status, wealth quintile, medical illness, pregnancy danger signs, experience of life-threatening events and household food insecurity were significantly associated with CAD. The likelihood of CAD were 84.0% (aOR = 0.16, 95% CI: 0.05, 0.56) lower among married women as compared to their counterparts. CAD was 2.8 times higher (aOR = 2.83, 95% CI: 1.17, 6.84) among participants in the highest wealth quintile as compared to participants in the lower quintile. Co-existing medical illness increases the odds of CAD by 3.6 times (aOR = 3.56, 95% CI: 1.68, 7.54) as compared to participants who were free from medical problems. Those participants who encountered pregnancy danger signs were 3 times (AOR = 2.66, (95% CI: 1.06, 6.67) higher risk to experience CAD as compared to their counterparts. The odds of CAD were 2 times (AOR = 2.11 (1.15, 3.92) higher among participants who experienced life-threatening events in the past six months. Likewise, the odds of CAD were 3.5 times (AOR = 3.51, 95% CI: 1.85, 6.64) higher among household food insecure participants (**Table 3**).

## Discussion

In our study, 10.04% of the pregnant women had antenatal CAD. High socioeconomic status, medical illness, pregnancy danger signs, life-threatening events and household food insecurity

**Table 1. Socio-demographic characteristics of participants in Arba Minch Zuria district, Gamo zone, southern Ethiopia 2019, (n = 667).**

| Variables | Category | Frequency | Percentage |
|---|---|---|---|
| Age in completed years | < 20 | 52 | 7.8 |
| | 20–24 | 161 | 24.1 |
| | 25–29 | 197 | 29.6 |
| | 30–34 | 161 | 24.1 |
| | ≥ 35 | 96 | 14.4 |
| Ethnicity | Gamo | 552 | 82.8 |
| | Zeyisie | 65 | 9.7 |
| | Wolayita | 42 | 6.3 |
| | [a]Other | 8 | 1.2 |
| Religion | Protestant | 480 | 72.0 |
| | Orthodox | 184 | 27.6 |
| | [b]Other | 3 | 0.4 |
| Marital status | Single | 14 | 2.1 |
| | Married | 646 | 96.9 |
| | [c]Other | 7 | 1.0 |
| Mother's education | No formal education | 360 | 54.0 |
| | Primary education (1–8) | 218 | 32.7 |
| | Secondary education (9–12) | 70 | 10.5 |
| | College and above | 19 | 2.8 |
| Husband's education | No formal education | 304 | 45.6 |
| | Primary education (1–8) | 222 | 33.3 |
| | Secondary education (9–12) | 87 | 13.0 |
| | College and above | 37 | 5.5 |
| | Missing | 17 | 2.5 |
| Mother's occupation | Farmer | 93 | 14.0 |
| | Housewife | 533 | 79.9 |
| | Trader | 28 | 4.2 |
| | [d]Other | 13 | 1.9 |
| Husband's occupation | Farmer | 439 | 65.8 |
| | Daily laborer | 118 | 17.7 |
| | Trader | 48 | 7.2 |
| | Government employ | 20 | 3.0 |
| | [e]Other | 25 | 3.7 |
| | Missing | 17 | 2.5 |
| Residence | Urban | 17 | 2.5 |
| | Rural | 650 | 97.5 |
| Family size | ≤ 5 | 414 | 62.1 |
| | > 5 | 253 | 37.9 |
| Wealth quintile | Low | 221 | 33.1 |
| | Medium | 225 | 33.7 |
| | High | 221 | 33.1 |

[a]Gofa, Gurag, Oromo.

[b]Muslim, Yewuha miskir.

[c]Divorced, widowed, married but not living together.

[d]Government employ, non-governmental employer, student.

[e]Unemployed, driver, non-governmental employer, student, a prophet.

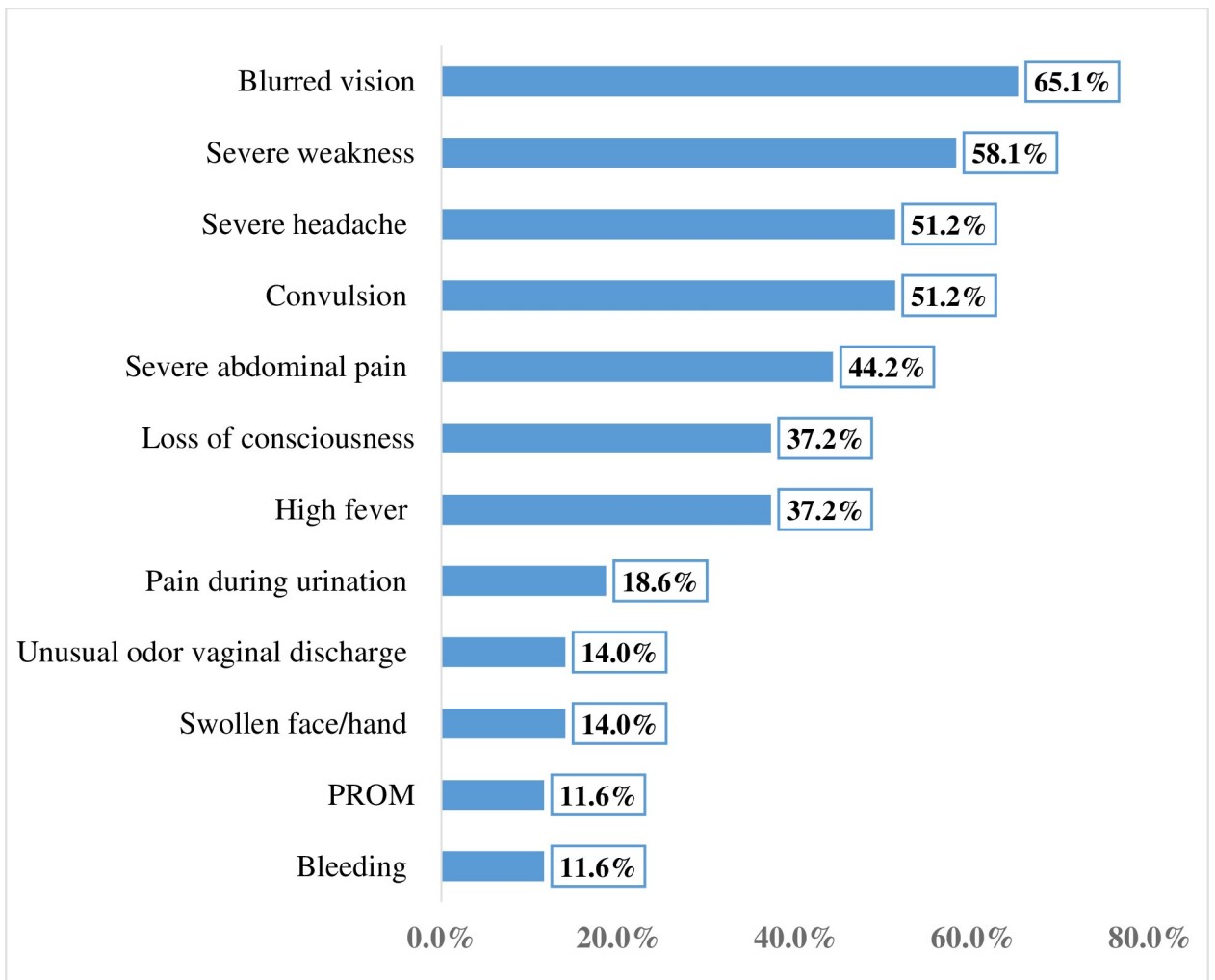

**Fig 1. Experience of pregnancy danger signs among women in Arba Minch zuria district, Gamo zone, southern Ethiopia, 2019 (n = 43).**

increased the odds of CAD. This finding is in line with the studies conducted in Pakistan 13.5% [6], Spain (9.5%) [7], South Africa (7.6% early and 9.7% in late pregnancy) [9] and a meta-analysis conducted by Falah and coworkers (9.5%) [4]. On the other hand, this finding is higher than the study conducted in Singapore (5.0%) [5] and very much lower than the study in Turkey (47.6%) [7]. These discrepancies might be due to differences in anxiety and depression measurement scales and cut-off point used, geographical settings, pregnancy trimester, socio-economic status and socio-cultural differences.

The likelihood of developing CAD during pregnancy was 84.0% lower among married women as compared to their counterparts. This finding is in agreement with the study conducted in Tanzania, where CAD was 4.5 times higher among single [40]. The possible justification is that presence of a husband with pregnant mother increases self-esteem, decreases stress and improves psychological wellbeing [52]. Furthermore, getting pregnant without formal marriage is strongly stigmatized by the community that results in stress and abortion, which further exposes the mother to depression and anxiety [53].

The odds of CAD were about three times higher among participants ranked in the highest wealth quintile. This is contrary to previous studies, where low socio-economic status is a risk

**Table 2. Obstetrics characteristics of participants in Arba Minch Zuria district, Gamo zone, southern Ethiopia, 2019, (n = 667).**

| Variables | Category | Frequency | Percentage |
|---|---|---|---|
| Age at first pregnancy | < 20 | 297 | 44.5 |
| | 20–24 | 315 | 47.2 |
| | ≥ 25 | 55 | 8.3 |
| Gravida | One | 121 | 18.2 |
| | Two to four | 363 | 55.0 |
| | Five and above | 179 | 26.8 |
| History of stillbirth | Yes | 26 | 3.9 |
| | No | 520 | 78.0 |
| | Missing | 121 | 18.1 |
| History of neonatal death | Yes | 72 | 10.8 |
| | No | 474 | 71.1 |
| | Missing | 121 | 18.1 |
| History of abortion | Yes | 47 | 7.0 |
| | No | 499 | 74.8 |
| | Missing | 121 | 18.1 |
| Gestational age in weeks | ≤ 12 | 46 | 6.9 |
| | 13–24 | 254 | 38.1 |
| | ≥ 25 | 367 | 55.0 |
| Reproductive intention | Planned | 622 | 93.3 |
| | Unplanned | 45 | 6.7 |
| ANC follow-up | Yes | 380 | 57.0 |
| | No | 287 | 43.0 |
| Place of ANC follow-up | Health post | 150 | 22.5 |
| | Health center | 208 | 31.2 |
| | Hospital | 20 | 3.0 |
| | Private clinic | 2 | 0.3 |
| | Missing | 287 | 43.0 |
| Number of ANC follow-up | < 4 | 259 | 38.8 |
| | ≥ 4 | 121 | 18.1 |
| | Missing | 287 | 43.0 |
| Experience of pregnancy danger signs | Yes | 43 | 6.4 |
| Medical illness | Yes | 59 | 8.9 |
| Intimate partner violence | Yes | 243 | 36.4 |
| Experience life-threatening events | Yes | 176 | 26.4 |
| Substance abuse | Yes | 51 | 7.7 |
| Household food security status | Secure | 479 | 72.1 |
| | Insecure | 186 | 27.9 |

factor for depression and anxiety [54, 55]. The possible justification is that economic status and necessity are parallel; struggle to attain more wealth leads to stress and tension. Also, in some cultures the household's resources are merely controlled by the husband, the woman always needs the permission of her husband to do anything despite the household wealth. Besides, when a man is rich enough the likelihood of polygyny (i.e. marriage between a man and multiple women) also increased; which further increases the odds of emotional abuse for the woman [56].

The odds of facing CAD were nearly four times higher among participants with co-existing medical illnesses. This is supported by previous studies, where the history of chronic medical

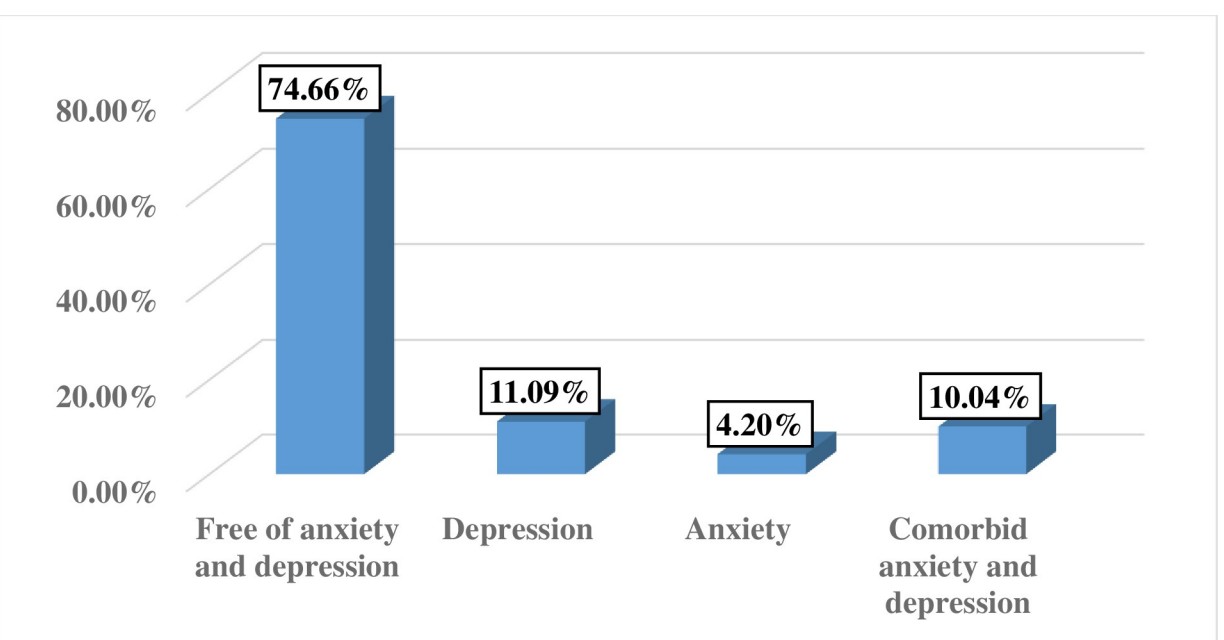

**Fig 2. Prevalence of common mental disorders among pregnant women in Arba Minch zuria district, Gamo zone, southern Ethiopia, 2019 (n = 667).**

illness increases the occurrence of mental disorders [11, 12]. Moreover, the presence of pregnancy danger signs increases the odds of CAD by threefold. This is consistent with a previous study in which the presence of obstetrics complications are associated with depression and anxiety [29]. This is probably due to increased frequency of health facility visits, incur extra cost and inhibition of the mother to lead a healthy life secondary to pregnancy-related ailments.

Those participants who experienced life-threatening events within the last six months were two times higher risk of encountering CAD. This finding is supported by previously published studies where adverse and stressful life events increase the likelihood of antenatal depression and anxiety [2, 57]. The possible justification is that stressful life events aggravate pregnancy-related complications including mental health problems.

Our study, also revealed that household food insecurity increases the odds of CAD by fourfold. Although no study exactly supports our finding, previously published studies consistently reported that household food insecurity increases the risk of perinatal common mental disorders [13, 14]. Besides, a study from South Africa also demonstrated that household food insecurity increases the odds of depression by fivefold; on the other hand, depression increases the odds of food insecurity by fourfold [15]. It is difficult to justify the exact mechanism by which household food insecurity results in depression and/or anxiety. But it might be because as the parents do not maintain their daily food requirements for them and their children they feel guilty and exposed to mental health problems.

In summary, the present study revealed that prenatal CAD is common in the study area. The study provides invaluable information for healthcare providers to address potential risk factors for CAD. Despite prenatal CAD results in serious health squeal both on the mother and her fetus, mental health services are very poor in Ethiopia [27]. Scholars suggested that mindfulness-based cognitive therapies (MBCT) are imperative to decrease depression and anxiety in pregnant women [58]. However, in Ethiopia, guidelines/manuals used to treat

**Table 3. Factors associated with comorbid anxiety and depression among pregnant women in Arba Minch Zuria district, Gamo zone, Southern Ethiopia, 2019, (n = 667).**

| Variables | [a]CAD | | [b]cOR (95% CI) | P-value | [c]aOR (95% CI) | P-value |
|---|---|---|---|---|---|---|
| | Yes (N, %) | No (N, %) | | | | |
| **Marital status** | | | | | | |
| Married | 60 (9.3) | 586 (90.7) | 0.20 (0.10, 0.53) | 0.001 | 0.16 (0.05, 0.56)** | 0.004 |
| Other[d] | 7 (33.3) | 14 (66.7) | 1 | | 1 | |
| **Intimate partner violence** | | | | | | |
| Yes | 42 (17.3) | 201 (82.7) | 3.33 (1.98, 5.63) | < 0.001 | 1.61 (0.88, 2.94) | 0.123 |
| No | 25 (5.9) | 399 (94.1) | 1 | | 1 | |
| **Wealth quintile** | | | | | | |
| Low | 9 (4.1) | 212 (95.9) | 1 | | 1 | |
| Medium | 28 (12.4) | 197 (87.6) | 3.35 (1.54, 7.27) | 0.002 | 1.54 (0.64, 3.69) | 0.333 |
| High | 30 (13.6) | 191 (86.4) | 3.70 (1.71, 7.99) | 0.001 | 2.83 (1.17, 6.84)* | 0.021 |
| **Medical Illness** | | | | | | |
| Yes | 22 (37.3) | 37 (62.7) | 7.4 (4.05, 13.67) | < 0.001 | 3.56 (1.68, 7.54)** | 0.001 |
| No | 45 (7.4) | 563 (92.6) | 1 | | 1 | |
| **Gestational age in weeks** | | | | | | |
| ≤12 | 2 (4.4) | 44 (95.6) | 1 | | 1 | |
| 13–24 | 20 (7.9) | 234 (92.1) | 1.9 (0.42, 8.33) | 0.406 | 1.62 (0.32, 8.10) | 0.559 |
| ≥25 | 45 (12.3) | 322 (87.7) | 3.1 (0.72, 13.12) | 0.129 | 2.40 (0.50, 11.58) | 0.276 |
| **Presence of danger signs in the current pregnancy** | | | | | | |
| Yes | 12 (27.9) | 31 (72.1) | 4.0 (1.95, 8.24) | < 0.001 | 2.66 (1.06, 6.67)* | 0.038 |
| No | 55 (8.8) | 569 (91.2) | 1 | | 1 | |
| **Social support** | | | | | | |
| Poor | 63 (11.6) | 480 (88.4) | 3.94 (1.4, 11.03) | 0.009 | 2.56 (0.80, 8.23) | 0.113 |
| Good | 4 (3.2) | 120 (96.8) | 1 | | 1 | |
| **Experience of life-threatening events** | | | | | | |
| Yes | 40 (22.7) | 136 (77.3) | 5.05 (2.99, 8.54) | < 0.001 | 2.11 (1.15, 3.92)* | 0.017 |
| No | 27 (5.5) | 464 (94.5) | 1 | | 1 | |
| **Substance abuse (alcohol)** | | | | | | |
| Yes | 10 (19.6) | 41 (80.4) | 2.39 (1.14, 5.03) | 0.021 | 1.96 (0.81, 4.75) | 0.137 |
| No | 57 (9.3) | 559 (90.7) | 1 | | 1 | |
| **Household food security status** | | | | | | |
| Secure | 25 (5.2) | 456 (94.8) | 1 | | 1 | |
| Insecure | 42 (22.6) | 144 (77.4) | 5.32 (3.13, 9.03) | < 0.001 | 3.51 (1.85, 6.64) ** | < 0.001 |

[a]Comorbid anxiety and depression.

[b]Crude Odds Ratio.

[c]Adjusted Odds Ratio.

[d]Divorced, widowed, married but not living together.

*Significant at P-value < 0.05.

**significant at P-value < 0.01.

pregnant women with depression/anxiety are not available and the healthcare providers working in the antenatal care units are not able to screen and manage these conditions. In low-income countries like Ethiopia with limited mental health centers and very low psychiatry professionals, any healthcare provider working in the antenatal care unit must be trained about common mental disorders screening and mindfulness interventions given for pregnant women. Besides, maternal mental health services must be integrated with other services like

reproductive, maternal and child health care services. Moreover, psychiatry professionals in collaboration with the federal ministry of health should develop easily accessible and understandable maternal mental health intervention guidelines.

Even though this study follows scientific methodology it might encounter the following limitations: social desirability bias might underestimate the response of some measurements such as IPV, substance abuse and household food insecurity. Moreover, the GAD-7 used to measure anxiety was not validated in the Ethiopian context; which might affect the result of the current study. Thus, researchers and psychiatry professionals must develop locally validated mental disorder screening tools.

## Conclusions

Generally, one in ten women encountered CAD in the study area. Being married, ranked in the highest wealth quintile, having medical illness, encountering pregnancy danger signs, experiencing life-threatening events and household food insecurity was significantly associated with CAD. Thus, incorporating psychological and psychosocial counseling as part of the routine ANC, early detection and treatment of pregnancy-related ailments and medical illness, and marriage counseling are pillars to diminish the onset of mental disorders during pregnancy.

## Supporting information

**S1 Table. Data collection tool.**
(DOCX)

**S2 Table. STORBE checklist.**
(DOC)

## Acknowledgments

The authors would like to acknowledge senior staffs in Arba Minch University, study participants, data collectors, supervisors and staff of AM-HDSS for their priceless support.

## Author Contributions

**Conceptualization:** Agegnehu Bante.

**Data curation:** Agegnehu Bante, Abera Mersha, Zerihun Zerdo, Biresaw Wassihun, Tomas Yeheyis.

**Formal analysis:** Agegnehu Bante, Abera Mersha.

**Funding acquisition:** Agegnehu Bante.

**Investigation:** Agegnehu Bante, Abera Mersha, Biresaw Wassihun, Tomas Yeheyis.

**Methodology:** Agegnehu Bante, Abera Mersha, Zerihun Zerdo, Biresaw Wassihun, Tomas Yeheyis.

**Project administration:** Agegnehu Bante, Abera Mersha, Zerihun Zerdo.

**Resources:** Agegnehu Bante, Zerihun Zerdo.

**Software:** Agegnehu Bante, Abera Mersha, Tomas Yeheyis.

**Supervision:** Agegnehu Bante, Abera Mersha, Zerihun Zerdo, Biresaw Wassihun, Tomas Yeheyis.

**Validation:** Agegnehu Bante, Tomas Yeheyis.

**Visualization:** Agegnehu Bante, Abera Mersha, Biresaw Wassihun.

**Writing – original draft:** Agegnehu Bante.

**Writing – review & editing:** Agegnehu Bante, Abera Mersha, Zerihun Zerdo, Biresaw Wassihun, Tomas Yeheyis.

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
