## [Decision Letter · Decision Letter 0]

11 Sep 2020

PONE-D-20-08361

Comorbid anxiety and depression: prevalence and associated factors among pregnant women in Arba Minch zuria district, southern Ethiopia

PLOS ONE

Dear Dr. Bante,

Thank you for submitting your manuscript to PLOS ONE. After careful consideration, we feel that it has merit but does not fully meet PLOS ONE’s publication criteria as it currently stands. Therefore, we invite you to submit a revised version of the manuscript that addresses the points raised during the review process.

**Editor's Comments:**

Line 15 - Suggest to avoid subjective and non-factual terms such as ‘earth-shattering’ in academic writing

Line 30 – in referring to the highest ranked quartile, please clarify if this means household income / socioeconomic level

Line 39 – While mental health during pregnancy is critically important, stating that it is ‘the major public health challenge in the world” might be contested particularly in light of the current global COVID-19 pandemic. Suggest to change tot “a major public health challenge”.

Line 42 – typo around topmost

Line 43 – please clarify what a ‘pocket study’ is

Line 67 -  consider moving MBCT to the Discussion

Line 87 – ? rationale for excluding women who were on treatment for anxiety / depression

Line 104 – what are pregnancy screen checks and how where they used in the sampling?

Line 128 – how many participants completed the pilot? It would be preferable to simply state the # rather than indicate 5% of the population

Line 162 – Ethics section could use more details such as whether compensation was provided, how exactly referrals for were done, if a participant was suicidal how was that managed?

Line 182 – the article would benefit from close proof reading for typos, example line 182 “got pregnant” rather than “get pregnant”

Results – Suggest to collapse some of the descriptive tables, not necessary to have a separate table for each major variable

Line 259 – explanation around the association between CAD and income status is not clear as written

Line 263 – does this mean that men take an additional wife when their first or earlier wives are pregnant?

Discussion – as currently written the Discussion is mostly a compare / contrast of the results with other existing studies. It would be improved by going deeper into what the results mean for women in Ethiopia, what policies/programs/services are in place for pregnant women with CAD, what should be in place, what are additional areas for future research, etc.?

The authors are also encouraged to use a cross-sectional study check list (ex. STROBE) to ensure that standards for reporting are met. 

We look forward to receiving your revised manuscript.

Kind regards,

Susan A. Bartels, MD, MPH, FRCPC

Academic Editor

PLOS ONE

Journal Requirements:

2. In the ethics statement in the Methods and online submission information, please clarify whether consent was written or verbal.  If verbal, please also specify: 1) whether the ethics committee approved the verbal consent procedure, 2) why written consent could not be obtained, and 3) how verbal consent was recorded. If your study included minors, state whether you obtained consent from parents or guardians. If the need for consent or parental consent was waived by the ethics committee, please include this information.

Reviewers' comments:

Reviewer's Responses to Questions

**Comments to the Author**

1. Is the manuscript technically sound, and do the data support the conclusions?

Reviewer #1: Partly

Reviewer #2: Yes

2. Has the statistical analysis been performed appropriately and rigorously? 

Reviewer #1: Yes

Reviewer #2: Yes

3. Have the authors made all data underlying the findings in their manuscript fully available?

Reviewer #1: Yes

Reviewer #2: Yes

4. Is the manuscript presented in an intelligible fashion and written in standard English?

Reviewer #1: No

Reviewer #2: Yes

5. Review Comments to the Author

Reviewer #1: Thank you very much for the opportunity to review this manuscript. Although the manuscript focused on an important area of research in Ethiopia, there are several issues which need to be addressed by the authors. Below are the concerns:

General comments:

1. There are so many grammatical errors throughout the manuscript. Language editing is recommended.

2. There are too many unscientific language use throughout the manuscript. For instance, “Despite anxiety and depression have an earth-shattering impact…”.

ABSTRACT:

1. The background needs to be re-written to clearly outline the rational for the present study.

2. The conclusion is too broad. It needs to be focused on the key findings from the study.

INTRODUCTION

1. The rationale for the present study is not clearly stated. The authors cited a study in Ethiopia by Woldetsadik et al. (2019) on prenatal mental health problems. How different is the current study in terms of added value. These issues should be clearly stated.

2. The authors need to provide more information on the choice of potential risk and protective factors by stating the theoretical bases for these factors.

3. Lines 39 & 40. The statements need to be edited and referenced as the authors are making factual claims.

METHODS

1. Exclusion and inclusion criteria need to be clearly spelt out.

2. Under “Sampling procedure”, the authors need to provide further details on “pregnancy screening checklist”.

3. The sections “Data collection tool” and “Study variables” should be put together. The authors need to provide detailed information on each study measure in terms of developers, number of items, response format and scoring as well as cut-offs.

4. The reliability values of the study instruments in the current study need to be stated.

5. The authors used “mindful interventions” throughout the manuscript. This needs to be corrected as it is “mindfulness interventions”.

6. Ethical clearance number needs to be provided if available.

RESULTS

1. Table 2. Education category “uneducated” should be changed to “no formal education”.

2. The results need to be summarised as the presentation of frequencies and percentages of each predictor variables makes the results to lengthy. I would suggest three key tables (Socio-demographic, Chi-Square and Logistic regression). You have the cut-offs or categorizations of each of these variables and therefore, there is no need for presenting all the items.

3. Check the spelling of ANXIETY in Fig 4.

DISCUSSION

1. The discussion is fairly written. However, results need to be situated in the Ethiopian context and not merely comparing with previous studies.

Reviewer #2: I think it is novel idea in antenatal women who have comorbid anxiety and depression which is public health challenge in low and middle income country. The paper is an important one that shows predictors of comorbid anxiety and depression, economical factors ,life events that encounter pregnant women, house hold food insecurity. One of the strength of this study also it community base study that taken from demographic surveillance site of Arbaminch university.

with regards

6. PLOS authors have the option to publish the peer review history of their article (what does this mean?). If published, this will include your full peer review and any attached files.

Reviewer #1: No

Reviewer #2: **Yes: **Wondale Getinet Alemu

---

## [Author Response · Author response to Decision Letter 0]

16 Sep 2020

Rebuttal Letter

September 13, 2020

Susan A. Bartels, MD, MPH, FRCPC

Academic Editor

PLOS ONE

Rebuttal Letter: Manuscript ID, PONE-D-20-08361

Dear Dr. Susan

Thank you for the opportunity to submit our manuscript “Comorbid anxiety and depression: prevalence and associated factors among pregnant women in Arba Minch zuria district, Gamo Zone, southern Ethiopia” for consideration for publication in PLOS ONE. We have carefully addressed the reviewers and editor comment in our responses and incorporated the related changes into the manuscript. To easily review our response: the comments and responses are written in black and dark blue font color respectively.

Thank you again for your consideration of this manuscript. 

Sincerely, 

Agegnehu Bante, MSc 

Arba Minch University Ethiopia 

Editor’s Comment #1

Line 15 - Suggest to avoid subjective and non-factual terms such as ‘earth-shattering’ in academic writing

Authors’ response: the introduction section of the abstract is revised and the term earth shattering is removed. Such errors are also identified in the rest of the document, amendments are done accordingly.

Editor’s Comment #2

Line 30 – in referring to the highest ranked quartile, please clarify if this means household income / socioeconomic level

Authors’ response: revised as ‘categorized in the highest socioeconomic level’

Editor’s Comment #3

Line 39 – While mental health during pregnancy is critically important, stating that it is ‘the major public health challenge in the world” might be contested particularly in light of the current global COVID-19 pandemic. Suggest to change tot “a major public health challenge”.

Authors’ response: since this study was conducted before covid-19 outbreak any misleading information like you mentioned are removed. The background section is modified including some citations and comments on this section are fully addressed. 

Editor’s Comment #4

Line 42 – typo around topmost

Authors’ response: that statement was not specific to pregnant women, hence it is total removed. 

Editor’s Comment #5

Line 43 – please clarify what a ‘pocket study’ is

Authors’ response: The phase was used just to indicate the study was conducted in one specific location (limited geographical area) of the country. However, to avoid any confusion for the readers the word pocket is removed in the revised file.

Editor’s Comment #6

Line 67 - consider moving MBCT to the Discussion

Authors’ response: the statement is take to the discussion section and paraphrased with other mindfulness interventions

Editor’s Comment #7

Line 87 – ? rationale for excluding women who were on treatment for anxiety / depression

Authors’ response: The purpose of the study was to determine magnitude of comorbid anxiety and depression. Thus, excluding participants with confirmed mental health problem and initiate therapy underestimates the problem. Fortunately, there is no participant who was excluded by this criteria. Hence, the eligibility criteria to enroll in the study was revised as follow: All pregnant women who were living in Arba Minch Health and Demographic Surveillance Site (AM-HDSS) were the study population. Women were enrolled in to the study irrespective of their age and pregnancy trimester. Those pregnant women who were not able to communicate due to serious illness and not available after three consecutive home visits were excluded from the study.

Editor’s Comment #8

Line 104 – what are pregnancy screen checks and how where they used in the sampling?

Authors’ response: The first step in the sampling procedure was identifying pregnant women in the selected cluster. The Arba Minch Health and Demographic Surveillance Site data base and the health extension workers registry was used to identify pregnant women. In addition, the WHO and CDC pregnancy assessment checklist; which is designed to determine whether the woman is pregnant or not just for family planning service, was used to reasonable sure the woman is pregnant. The detail of the questions and how they used are included under the data collection tool and uploaded as a supplementary file.

Editor’s Comment #9

Line 128 – how many participants completed the pilot? It would be preferable to simply state the # rather than indicate 5% of the population

Authors’ response: the minimum sample required for this study was calculated as 654. However, due to the nature of the cluster sampling a total of 676 pregnant women were identified in the selected cluster. Hence, 5% of 676; 34 pregnant women who lived in Arba Minch town were selected for the pretest and amendment was done based on the result.

Editor’s Comment #10

Line 162 – Ethics section could use more details such as whether compensation was provided, how exactly referrals for were done, if a participant was suicidal how was that managed?

Authors’ response: participation in the current study was fully voluntary and there is no compensation that directly given for the participant. Since, all participants were under the Arba Minch health demographic surveillance site those who were screened positive for depression/anxiety were easily identified by their household number in the database. In general the ethics statement is revised as follow: Ethical approval for the study protocol was provided by Arba Minch University, college of medicine and health sciences, Institutional Review Board (IRB) with an ethical clearance number of CMHS/12031653/111. Moreover, this study was conducted under the Declaration of Helsinki, and the participants were informed of the purpose of the study. Furthermore, before the commencement of data collection written informed consent was obtained from all participants. To maintain the confidentiality of information gathered from each study participant, code numbers were used throughout the study. Pregnant women who screed positive for depression or anxiety received onsite mindfulness interventions. In addition, they were linked with the health extension workers for regularly psychological interventions and referral to health facilities in case of sever case like suicidal ideation.

Editor’s Comment #11

Line 182 – the article would benefit from close proof reading for typos, example line 182 “got pregnant” rather than “get pregnant”

Authors’ response: the manuscript passes through careful proof reading by all authors; Grammar and editorial issues including the raised one are addressed 

Editor’s Comment #12

Results – Suggest to collapse some of the descriptive tables, not necessary to have a separate table for each major variable

Authors’ response: the result section is presented shortly; some extra details are removed and presented only in 3 tables and 3 figures with some text paraphrasing.

Editor’s Comment #13

Line 259 – explanation around the association between CAD and income status is not clear as written

Authors’ response: This might be because of economic status increased the need for the individual also increased and struggle to acquire more capital that leads to mental disorder. Also, in some cultures everything is controlled by the husband, the mother does nothing without the permission of her partner. Furthermore, as the wealth of the family improved the husband needs an extra wife, which is commonly exercised during pregnancy. 

Editor’s Comment #14

Line 263 – does this mean that men take an additional wife when their first or earlier wives are pregnant?

Authors’ response: the interpretation is revised as it placed in the preceding comment. But let me explain your concern: 

Although it is diminished, the culture allows two and more wives if the man is rich and able to fulfill the need of all his wives. However, in realty there is a conflict of interest between wives and the husband lives here and there such norms put tremendous impact in women. 

Editor’s Comment #15

Discussion – as currently written the Discussion is mostly a compare / contrast of the results with other existing studies. It would be improved by going deeper into what the results mean for women in Ethiopia, what policies/programs/services are in place for pregnant women with CAD, what should be in place, what are additional areas for future research, etc.?

Authors’ response: all the ideas raised here are added in the last two paragraphs of the discussion as follow: The present study revealed that CAD in pregnant women is common in the study area. The study provides invaluable information for healthcare providers to address potential risk factors for depression and anxiety. Despite prenatal CAD results in serious health consequence both for the mother and her fetus, mental health services are very poor in Ethiopia. Scholars suggested that mindfulness-based cognitive therapy (MBCT) has imperative to decrease depression and anxiety in pregnant women. However, in Ethiopia, guidelines/manuals used to treat pregnant women with depression/anxiety are not available and the healthcare providers working in the antenatal care units are not able to screen and manage these conditions. In low income countries like Ethiopia, limited mental health centers and very low psychiatry professional, healthcare providers working in the antenatal care unit must be trained about common mental disorders screening and mindfulness interventions given for pregnant women. Moreover, maternal mental health must be integrated with other services like reproductive health, maternal and child health care. Psychiatry professional in collaboration with the federal ministry health should develop maternal mental health intervention guidelines. 

Even though this study follows scientific methodology, it might encounter the following limitations: social desirability bias might underestimate the response of some measurements such as intimate partner violence substance abuse, household food insecurity. Moreover, the GAD-7 used to measure anxiety was not validated in Ethiopian context; which might affect the result of the current study. Thus, researchers and psychiatry professional must develop locally validated mental disorder screening tools.

Editor’s Comment #16

The authors are also encouraged to use a cross-sectional study check list (ex. STROBE) to ensure that standards for reporting are met. 

Authors’ response: the STORBE-checklist is filled and uploaded as a supplementary information in the system

Editor’s Comment #17

https://journals.plos.org/plosone/s/file?id=wjVg/PLOSOne_formatting_sample_main_body.pdf &https://journals.plos.org/plosone/s/file?id=ba62/PLOSOne_formatting_sample_title_authors_affiliations.pdf

Authors’ response: the revised manuscript meets all the journal style requirements; 

Editor’s Comment #18

2. In the ethics statement in the Methods and online submission information, please clarify whether consent was written or verbal. If verbal, please also specify: 1) whether the ethics committee approved the verbal consent procedure, 2) why written consent could not be obtained, and 3) how verbal consent was recorded. If your study included minors, state whether you obtained consent from parents or guardians. If the need for consent or parental consent was waived by the ethics committee, please include this information.

Authors’ response: the consent was written and it is modified both in the method section and online submission. Let us explain more about the ethical statement: 

The participant information sheet and informed voluntary consent was prepared in English and Amharic language. The information sheet mainly includes the title, purpose of the study, procedure and duration, risk and benefits, confidentiality, rights and contact address for any query. In addition, there is declaration of informed voluntary consent statement that was given for the participant for signature; it states that “I have read/was read to me the participant information sheet. I have clearly understood the purpose of the study, the procedures, the risks and benefits, issues of confidentiality, the rights of participating and contact address for any queries. I have been given the opportunity to ask questions for things that may have been unclear. I was informed that I have the right to withdraw from the study at any time or not to answer any question that I do not want. Therefore, I declare my voluntary consent to participate in this study with my initials (signature).” Furthermore, the proposal received an ethical clearance letter from Arba Minch University, college of medicine and health sciences, Institutional Review Board (IRB) with an ethical clearance number of CMHS/12031653/111.

Reviewers comment

Reviewer #1: Thank you very much for the opportunity to review this manuscript. Although the manuscript focused on an important area of research in Ethiopia, there are several issues which need to be addressed by the authors. Below are the concerns: 

General comments:

Reviewer #1 comment #1

1. There are so many grammatical errors throughout the manuscript. Language editing is recommended.

Authors’ response: as mentioned above the manuscript pass through proof reading by all authors for many times. In addition, grammarly free application was used to avoid some spelling and grammar errors. 

Reviewer #1 comment #2

2. There are too many unscientific language use throughout the manuscript. For instance, “Despite anxiety and depression have an earth-shattering impact…”.

Authors’ response: all the authors go through the whole document and replace unscientific/ unclear words with simple and easily understandable words. 

ABSTRACT:

Reviewer #1 comment #3

1. The background needs to be re-written to clearly outline the rational for the present study.

Authors’ response: based on the suggestion the background section is revised by incorporating more recent articles and the rational of the study is explained by showing the research gap. Even though the prevalence of depression in pregnant women is studied, as far as our knowledge studies that determines the prevalence of comorbid depression and anxiety; anxiety alone are not researched. Hence, to design comprehensive mental disorder interventions the current study has paramount contribution. In the revise manuscript the rational of the study is incorporated.

Reviewer #1 comment #4

2. The conclusion is too broad. It needs to be focused on the key findings from the study.

Authors’ response: the conclusion written based on the key findings and revised as: In general, one in every ten women were affected by CAD. Early identification and treatment of pregnancy dangers signs and medical illness, incorporating mindfulness interventions as part of the routine antenatal care and improving household food security have paramount importance to minimize prenatal mental health problems.

INTRODUCTION

Reviewer #1 comment #5

1. The rationale for the present study is not clearly stated. The authors cited a study in Ethiopia by Woldetsadik et al. (2019) on prenatal mental health problems. How different is the current study in terms of added value. These issues should be clearly stated.

Authors’ response: The study conducted by Woldetsadik et al. (2019) in Bale zone South-East Ethiopia mainly focused on depression alone despite the running title is about “Prevalence of common mental disorder and associated factors among pregnant women”. In addition the measurement scale used to assess depression and anxiety is totally different. In our study the patient health questioner (PHQ); which has good internal consistency and validated in Ethiopian context was used for the antenatal depression assessment. Likewise, antenatal anxiety was assessed using the Generalised Anxiety Disorder (GAD-7); which is a useful and a validated tool to assess perinatal anxiety. On the contrary, the study conducted by Woldetsadik et al. (2019) used the WHO Self-Reported Questionnaire (SRQ); which is mainly used to assess depression. The paper was cited just to show the previous studies conducted in Ethiopia are manly depression alone. Based on the suggestion, the citation is removed from the background and discussion section it is not possible to compare the depression alone with that of co-morbid depression and anxiety. 

Reviewer #1 comment #6

2. The authors need to provide more information on the choice of potential risk and protective factors by stating the theoretical bases for these factors.

Authors’ response: after extensive review of literature all variables associated with the co-morbid anxiety and depression. In the revised version these variables are in a separate paragraph. In the main document the conceptual framework and the relationship of each variable was clearly explained. However, presenting the theoretical bases for each variable make the background bulky and too long. Therefore, factors associated with CAD are presented in short way. 

 Reviewer #1 comment #7

3. Lines 39 & 40. The statements need to be edited and referenced as the authors are making factual claims.

Authors’ response: as per the suggestion; the statement is rewritten and cited. In addition, some of the statement are replaced with more recent data. 

METHODS

Reviewer #1 comment #8

1. Exclusion and inclusion criteria need to be clearly spelt out.

Authors’ response: excluding pregnant women who initiate treatment for mental health problems underestimates the magnitude of the problem and in the revised manuscript the eligibility criteria for the study is modified as: All pregnant women who were living in Arba Minch Health and Demographic Surveillance Site (AM-HDSS) were the study population. Women were enrolled in to the study irrespective of their age and pregnancy trimester. Those pregnant women who were not able to communicate due to serious illness and not available after three consecutive home visits were excluded from the study.

Reviewer #1 comment #9

2. Under “Sampling procedure”, the authors need to provide further details on “pregnancy screening checklist”.

Authors’ response: the first step of the data assortment process was identified pregnant women from the selected clusters. Thus, the Arba Minch Health and Demographic Surveillance and the health extension workers pregnancy register were used to identify pregnant women in each cluster. In addition, to minimize the chance of missing early pregnancies the WHO and CDC pregnancy screening checklist was used to reasonable determine the pregnancy status of the woman (Table 1). 

Table 1: Checklist used to determine the Pregnancy Status of women in Arba Minch zuria, Gamo zone, southern Ethiopia, 2019

Questions Response

1. Did your last menstrual period start within the last 7 day? [1] Yes [2] No

2. Have you had sex since your last menstrual period started? [1] Yes [2] No

3. Have you been using a reliable birth control method consistently and correctly since your last menstrual period started? [1] Yes [2] No

4. Did you have a miscarriage or abortion in the last 7 day? [1] Yes [2] No

5. Did you have a baby in the last 4 week? [1] Yes [2] No

6. Did you have a baby in the last 6 month?

a. Are you breastfeeding now? 

b. Are your periods still gone since you had the baby? [1] Yes [2] No

[1] Yes [2] No

[1] Yes [2] No

*Pregnancy was considered to be unlikely if the participant fulfilled any of these criteria: answered “Yes” to questions 1, 3, 4, or 5; answered “no” to question 2; or answered “Yes” to questions 6, 6a, and 6b. The health extension workers were instructed to pass the household if there is no pregnant woman after assessing using the above questions.

Reviewer #1 comment #10

3. The sections “Data collection tool” and “Study variables” should be put together. The authors need to provide detailed information on each study measure in terms of developers, number of items, response format and scoring as well as cut-offs.

Authors’ response: the study variables section is merged to the “data collection tool” section and the detail of each variables measurement are described as follow: 

Outcome variables measurement scales and their measurement 

Depression: The Patient Health Questionnaire (PHQ-9) is originally designed by Doctors Spitzer, Williams and Kroenke was used for depression assessment. PHQ-9 score range from 0 to 27, since each of the 9 items can be scored from 0 (not at all) to 3 (nearly every day) [1]. Based on this severity of can be categorized as: 0-4 none, 5-9 mild, 10-14 moderate, 15-19 moderately severe, 20-27 severe. In our case, depression was categorized in to two; a score of ≥ 5 was used as a cutoff point to categorize pregnant women as having depression because there were small observation under moderate (13 observations) and moderately severe (6 observations) depression category and no observation under severe depression. Hence, the mild, moderate and moderately severe depression were merged as one category and labeled as “depression” [2]. 

Anxiety: The Generalized Anxiety Disorder 7-item (GAD-7) scale, developed by Robert L. Spitzer, MD, served as professor of psychiatry at Columbia University for 49 years, was used for anxiety assessment. GAD-7 is a valid and efficient tool for screening for generalized anxiety disorder in clinical practice and research [3]. GAD-7 score range from 0 to 21, since each of the 7 items can be scored from 0 (not at all) to 3 (nearly every day). Scores of 5, 10, and 15 are taken as the cut-off points for mild, moderate and severe anxiety, respectively. In our study score of ≥5 was taken as the cutoff point to categorize pregnant women as having anxiety [4]; there is no observation with a score of ≥15 (sever) and only 19 observation with a score of 10-14 (moderate). 

Independent variables and measurements (i.e. composite variables)

Substance abuse: it was assessed using Fast Alcohol Screening Test (FAST) scale developed from the Alcohol Use Disorders Identification Test (AUDIT) by Hodgson et al. 2002. FAST has 4 item with a score ranges from 0 to 16, where a score of ≥3 indicates harmful drinking [5]. 

Intimate partner violence (IPV): it was assessed using five item; Women's Abuse Screening Test (WAST) originally developed by Brown et al. 1996 [6] was used. WAST score ranges from 0-16 where a score > 1 indicates the presence of violence [7].

Social Support: Oslo Social Support Scale (OSSS-3) which has three item scale and originally developed in Neverland’s was used. Its score ranges from 3-14 and categorized into three as: “poor social support” 3-8, “moderate support” 9-11 and “strong support” 12-14 [8]. 

Life events: The List of Threatening Experiences (LTE) developed by Brown & Harris (1978) was used; LTE is self-reported questionnaire with a list of 12 threatening events that invite a dichotomous response (yes/no). The number of life events were counted for each respondent and recorded as a value ranging from 0 to 12 [9]. LTE has good test-retest reliability and internal

consistency and widely used across the globe.

Food security status: it was measured using the Household Food Insecurity Access Scale (HFIAS) developed by Food and Nutrition Technical Assistance (FANTA) Program of the U.S. Agency for International Development. It has 9 items with a follow-up question about the frequency of occurrence if the response is yes for the main questions. The score range from 0 to 27. The HFIAS category was calculated for each household using the following command: 

Food Secure: HFIA category = 1 if [(Q1a=0 or Q1a=1) and Q2=0 and Q3=0 and Q4=0 and Q5=0 and Q6=0 and Q7=0 and Q8=0 and Q9=0]

Mildly Food Insecure Access: HFIA category = 2 if [(Q1a=2 or Q1a=3 or Q2a=1 or Q2a=2 or Q2a=3 or Q3a=1 or Q4a=1) and Q5=0 and Q6=0 and Q7=0 and Q8=0 and Q9=0]

Moderately Food Insecure Access: HFIA category = 3 if [(Q3a=2 or Q3a=3 or Q4a=2 or Q4a=3 or Q5a=1 or Q5a=2 or Q6a=1 or Q6a=2) and Q7=0 and Q8=0 and Q9=0]

Severely Food Insecure Access: HFIA category = 4 if [Q5a=3 or Q6a=3 or Q7a=1 or Q7a=2 or Q7a=3 or Q8a=1 or Q8a=2 or Q8a=3 or Q9a=1 or Q9a=2 or Q9a=3]

In the current study, for the multivariable analysis household food security was categorized in to two as “food secure” when the participants didn’t experience food insecurity (access) conditions in the past 4 weeks and “food insecure” when the unable to access sufficient food at all time to lead active and healthy life [10]. 

Reviewer #1 comment #11

4. The reliability values of the study instruments in the current study need to be stated.

Authors’ response: the reliability of the PHQ-9 and GAD-7 were calculated for this particular study and the Cronbach’s alpha for the PHQ-9 = 87 and for the GAD-7=0.83

Table 2: Stata output for the reliability values for the PHQ-9 and GAD-7 

PHQ-9 GAD-7

Command 

alpha ltlintrst-ththurt

Output 

Test scale = mean(unstandardized items)

Average interitem covariance: .1030973

Number of items in the scale: 9

Scale reliability coefficient: 0.8676 Command 

alpha flntvos-afraid

Output 

Test scale = mean(unstandardized items)

Average interitem covariance: .1175975

Number of items in the scale: 7

Scale reliability coefficient: 0.8322

Reviewer #1 comment #12

5. The authors used “mindful interventions” throughout the manuscript. This needs to be corrected as it is “mindfulness interventions”.

Authors’ response: checked throughout the document and amendment was done accordingly

Reviewer #1 comment #13

6. Ethical clearance number needs to be provided if available.

Authors’ response: yes it is available, “CMHS/12031653/111”, and incorporated in the revised manuscript 

RESULTS

Reviewer #1 comment #14

1. Table 2. Education category “uneducated” should be changed to “no formal education”.

Authors’ response: “uneducated” replaced with “No formal education”

Reviewer #1 comment #15

2. The results need to be summarised as the presentation of frequencies and percentages of each predictor variables makes the results to lengthy. I would suggest three key tables (Socio-demographic, Chi-Square and Logistic regression). You have the cut-offs or categorizations of each of these variables and therefore, there is no need for presenting all the items.

Authors’ response: some of the table are removed only the socio-demographic, obstetrics and the logistic regression table are presented. The chi-square is already incorporated under the logistic regression hence no need of presenting it separately. 

Reviewer #1 comment #16

3. Check the spelling of ANXIETY in Fig 4.

Authors’ response: yes it is checked and revised; the revised figure is uploaded to the system 

DISCUSSION

Reviewer #1 comment #17

1. The discussion is fairly written. However, results need to be situated in the Ethiopian context and not merely comparing with previous studies.

Authors’ response: as mentioned under the editor’s comment section the discussion mainly shows the existing interventions for maternal mental disorders and forwards future direction both for researchers, psychiatry professional and other concerned stakeholders: key points added under the discussion section are presented as follow: 

Despite prenatal CAD results in serious health consequence both for the mother and her fetus, mental health services are very poor in Ethiopia. Scholars suggested that mindfulness-based cognitive therapy (MBCT) has imperative to decrease depression and anxiety in pregnant women. However, in Ethiopia, guidelines/manuals used to treat pregnant women with depression/anxiety are not available and the healthcare providers working in the antenatal care units are not able to screen and manage these conditions. In low income countries like Ethiopia, limited mental health centers and very low psychiatry professional, healthcare providers working in the antenatal care unit must be trained about common mental disorders screening and mindfulness interventions given for pregnant women. Moreover, maternal mental health must be integrated with other services like reproductive health, maternal and child health care. Psychiatry professional in collaboration with the federal ministry health should develop maternal mental health intervention guidelines.

Reviewer #2 comment

I think it is novel idea in antenatal women who have comorbid anxiety and depression which is public health challenge in low and middle income country. The paper is an important one that shows predictors of comorbid anxiety and depression, economical factors, life events that encounter pregnant women, house hold food insecurity. One of the strength of this study also it community base study that taken from demographic surveillance site of Arba Minch university.

Authors’ response: Thank You!!

References 

1. Kroenke, K., R.L. Spitzer, and J.B.J.J.o.g.i.m. Williams, The PHQ‐9: validity of a brief depression severity measure. 2001. 16(9): p. 606-613.

2. Bitew, T., et al., Antenatal depressive symptoms and maternal health care utilisation: a population-based study of pregnant women in Ethiopia. BMC Pregnancy Childbirth, 2016. 16(1): p. 301.

3. Spitzer, R.L., et al., A brief measure for assessing generalized anxiety disorder: the GAD-7. 2006. 166(10): p. 1092-1097.

4. Zhong, Q.-Y., et al., Diagnostic validity of the Generalized Anxiety Disorder-7 (GAD-7) among pregnant women. PloS one, 2015. 10(4).

5. Hodgson, R., et al., The FAST alcohol screening test. 2002. 37(1): p. 61-66.

6. Brown, J.B., et al., Development of the Woman Abuse Screening Tool for use in family practice. 1996. 28: p. 422-428.

7. Rabin, R., et al., Intimate partner violence screening tools. American Journal of Prev Med, 2009. 36(5): p. 439.

8. Boen, H., Characteristics of senior centre users – and the impact of a group programme on social support and late-life depression. Norsk Epidemiologi 2012. 22(2): p. 261.

9. Brugha, T. and D. Cragg, The List of Threatening Experiences: the reliability and validity of a brief life events questionnaire. Acta Psychiatr Scand, 1990. 82(1): p. 77.

10. Coates, J., A. Swindale, and P. Bilinsky, Household Food Insecurity Access Scale (HFIAS) for measurement of food access: indicator guide. Washington, DC: food and nutrition technical assistance project, academy for educational Development, 34. 2007.

---

## [Decision Letter · Decision Letter 1]

3 Feb 2021

PONE-D-20-08361R1

Comorbid anxiety and depression: prevalence and associated factors among pregnant women in Arba Minch zuria district, Gamo zone, southern Ethiopia

PLOS ONE

Dear Dr. Bante,

Thank you for submitting your manuscript to PLOS ONE. After careful consideration, we feel that it has merit but does not fully meet PLOS ONE’s publication criteria as it currently stands. Therefore, we invite you to submit a revised version of the manuscript that addresses the points raised during the review process.

We suggest you thoroughly copyedit your manuscript for language usage, spelling, and grammar. If you do not know anyone who can help you do this, you may wish to consider employing a professional scientific editing service.

We look forward to receiving your revised manuscript.

Kind regards,

Susan A. Bartels, MD, MPH, FRCPC

Academic Editor

PLOS ONE

Reviewers' comments:

Reviewer's Responses to Questions

**Comments to the Author**

1. If the authors have adequately addressed your comments raised in a previous round of review and you feel that this manuscript is now acceptable for publication, you may indicate that here to bypass the “Comments to the Author” section, enter your conflict of interest statement in the “Confidential to Editor” section, and submit your "Accept" recommendation.

Reviewer #3: (No Response)

Reviewer #4: All comments have been addressed

Reviewer #5: All comments have been addressed

2. Is the manuscript technically sound, and do the data support the conclusions?

Reviewer #3: Partly

Reviewer #4: (No Response)

Reviewer #5: Yes

3. Has the statistical analysis been performed appropriately and rigorously? 

Reviewer #3: I Don't Know

Reviewer #4: (No Response)

Reviewer #5: Yes

4. Have the authors made all data underlying the findings in their manuscript fully available?

Reviewer #3: No

Reviewer #4: (No Response)

Reviewer #5: Yes

5. Is the manuscript presented in an intelligible fashion and written in standard English?

Reviewer #3: No

Reviewer #4: (No Response)

Reviewer #5: Yes

6. Review Comments to the Author

Reviewer #3: Reviewer's comments PONE-D-20-08361R1

Thank you for the opportunity to review this manuscript. This manuscript describes the prevalence of and factors associated with comorbid depression and anxiety during pregnancy in a community sample, in Ethiopia. Rates of both antenatal depression and anxiety have been shown to be concerningly high in other African studies, and both have negative consequences for mother and offspring. As the authors mention, there is a dearth of research on the comorbidity of these conditions, and therefore this is an important piece of work.

General comments

1. I feel that it would be beneficial to have this manuscript revised by a copy editor as there are numerous sentences which are confusing. An example is Line 44- 45: Depression is frequently occurs with anxiety and results an enormous adverse pregnancy and birth outcomes.

Abstract

1. I do not feel that the first reviewer’s concerns about the abstract have been addressed. There needs to be a stronger rationale for the current study, and the conclusion needs to speak to the results. The current conclusion reads as though a mindfulness intervention for improving mental health was undertaken alongside improving household food security.

Figures and tables

2. Please show all categories in Tables 1 and 2, and include a category for ‘missing’ data in instances where there is only data for some of the n=667 participants. Also, percentages should be calculated on the full analytic sample. For example, in Table 1, Husband’s education should be displayed as follows:

No formal education 304 45.6

Primary education 222 33.3

Secondary education 87 13.0

College and above 37 5.5

Missing 17 2.5

3. In table 2, “Status of current pregnancy” could be called “reproductive intention”

4. I am not convinced that Fig 2 is necessary.

5. Table 3: I find it confusing that the reference group is not listed first in this table – I have not seen results presented in this way before but could be mistaken.

6. I think it would be useful to see the p-value for the multivariate regression models presented in table 3.

Introduction

2. I do not feel that the prevalence of mental disorders in LMIC/ African antenatal populations was adequately discussed in the introduction, and noticed that there were African studies of mental health (including Ethiopian) which was not considered / cited. Authors cite Ethiopian studies in Line 66-68 – it would be interesting to know prevalence’s found in these studies for context and comparison.

I realise that the literature on comorbid depression and anxiety is sparse, but there are some studies which may be useful:

• Sawyer A, Ayers S, Smith H. Pre- and postnatal psychological wellbeing in Africa: A systematic review. Journal of Affective Disorders 2010; 123(1–3): 17-29.

• Baron EC, Hanlon C, Mall S, et al. Maternal mental health in primary care in five low- and middle-income countries: a situational analysis. BMC health services research 2016; 16: 53.

• Redinger S, Pearson RM, Houle B, Norris SA, Rochat TJ. Antenatal depression and anxiety across pregnancy in urban South Africa. Journal of affective disorders 2020; 277: 296-305.

• Biaggi A, Conroy S, Pawlby S, Pariante CM. Identifying the women at risk of antenatal anxiety and depression: A systematic review. Journal of Affective Disorders 2016; 191: 62-77.

• (for medical condition comparison) Sowa NA, Cholera R, Pence BW, Gaynes BN. Perinatal depression in HIV-infected African women: a systematic review. J Clin Psychiatry 2015; 76(10): 1385-96.

3. Line 45: Given the wide disparity in prevalences of depression and anxiety stated in the introduction, it would be help to have more information about the studies you cite when describing rates of comorbidity – for example were participants from HIC/LMIC or low/high risk pregnancies etc.

There is an increasing trend in comorbid anxiety and depression (CAD); Falah et.al. reported that 9.5% of 25,592 participants had self-reported 46 antenatal anxiety and depressive symptoms

4. Lines 50-55 explains the factors associated with “comorbid anxiety and depression” in the literature however only the bulk of cited literature is for either depression or anxiety. Might be useful to describe factors associated with one or the other in Africa/ Ethiopia specifically (as they are slightly different) and then tell the reader what is associated with comorbidity (this literature is sparse, might need to draw from HIC literature).

Methods

5. When using the PHQ-9 as a measure of probable depression, a cut-off score of 10 is what has been shown in validation studies to represent a depression, including in antenatal populations. I know that a cutoff of >5 has been used in Ethiopia before, and I assume the authors have based their analysis on this however the reasoning given by the authors in the manuscript is confusing and makes it seem like prevalence’s are being under-estimated. I would reconsider this explanation. The same is true for the description of the GAD-7 cutoff and analysis.

Line score of ≥5 was used as a cut-off point for possible antenatal depression [23] due to small/null observations under moderate and moderately severe and severe depression.

Results

6. Lines 229 -231 Please state what the confounding factors in the model were?

7. Normal convention for reporting regression results: AOR 95% CI [CI-CI] p-value

8. The fact that women in the highest socioeconomic bracket had higher odds of having comorbid depression and anxiety warrants a more detailed explanation / discussion – especially since women reporting food insecurity were at higher risk. It would be interesting to look at/ mention what the unadjusted results were for this variable to see if they were in the same direction, or if there was collinearity with the food insecurity variable in the multivariate regression.

Discussion

9. Lines 260 and 261 – consider referencing.

10. Generally I echo the comments of the previous reviewer on the discussion section – that it is fairly written but would benefit from situating results within the LMIC then African then Ethiopian context.

11. Nice papers to look at when considering the results of food insecurity:

• Trujillo J, Vieira MC, Lepsch J, et al. A systematic review of the associations between maternal nutritional biomarkers and depression and/or anxiety during pregnancy and postpartum. Journal of Affective Disorders 2018; 232: 185-203.

• Abrahams Z, Lund C, Field S, Honikman S. Factors associated with household food insecurity and depression in pregnant South African women from a low socio-economic setting: a cross-sectional study. Social psychiatry and psychiatric epidemiology 2018; 53(4): 363-72.

Reviewer #4: In my opinion, the authors have addressed all the comments and suggestions made by the reviewers and the manuscript could be published

Reviewer #5: (No Response)

7. PLOS authors have the option to publish the peer review history of their article (what does this mean?). If published, this will include your full peer review and any attached files.

Reviewer #3: No

Reviewer #4: No

Reviewer #5: No

---

## [Author Response · Author response to Decision Letter 1]

10 Feb 2021

Rebuttal Letter

February 10, 2021

Susan A. Bartels, MD, MPH, FRCPC

Academic Editor

PLOS ONE

Rebuttal Letter: Manuscript ID, PONE-D-20-08361

Dear Dr. Susan

Thank you for the opportunity to submit our manuscript “Comorbid anxiety and depression: prevalence and associated factors among pregnant women in Arba Minch zuria district, Gamo Zone, southern Ethiopia” for consideration for publication in PLOS ONE. We have carefully addressed the reviewers' and editor's comments in our responses and incorporated the related changes into the manuscript. To easily review our response: the comments and responses are written in black and dark blue font color respectively.

Thank you again for your consideration of this manuscript. 

Sincerely, 

Agegnehu Bante, MSc in maternity and neonatal nursing 

Arba Minch University, Ethiopia 

Reviewer's comments 

Reviewer #3

Thank you for the opportunity to review this manuscript. This manuscript describes the prevalence of and factors associated with comorbid depression and anxiety during pregnancy in a community sample, in Ethiopia. Rates of both antenatal depression and anxiety have been shown to be concerningly high in other African studies, and both have negative consequences for mother and offspring. As the authors mention, there is a dearth of research on the comorbidity of these conditions, and therefore this is an important piece of work. 

General 

1. I feel that it would be beneficial to have this manuscript revised by a copy editor as there are numerous sentences which are confusing. An example: 

Line 44- 45: Depression is frequently occurs with anxiety and results an enormous adverse pregnancy and birth outcomes. 

Authors’ response: after addressing all the constructive comments you raised, the final version of the manuscript go through proof reads by the corresponding author, co-authors and senior scholars. Free online article paraphrasing too, plagiarism checker and grammarly was used for grammatical and spelling errors. Besides, the manuscript is written in simple and easily understandable English. Thus, in the current version all misleading, and unclear information is written understandably. 

2. Figures and tables 

Please show all categories in Tables 1 and 2, and include a category for ‘missing’ data in instances where there is only data for some of the n=667 participants. Also, percentages should be calculated on the full analytic sample. For example, in Table 1, Husband’s education should be displayed as follows: 

No formal education 304 45.6 

Primary education 222 33.3 

Secondary education 87 13.0 

College and above 37 5.5 

Missing 17 2.5 

 Authors’ response: In our case, it is not missing rather skip questions. For example, in table 1 ‘husband education’ and ‘husband occupation’ were not asked from single, divorced and widowed mothers that is why the denominator is 650 and the valid percentage is calculated based on this value. Likewise, in Table 2: the percentage for ‘history of stillbirth’, history of neonatal death, history of abortion’ were calculated by taking the denominator 546 because primigravida mothers were not asked these questions. Furthermore, “the number of ANC visits and Place of ANC follow-up” were asked only from those who already initiate ANC follow-up (i.e. 380 participants). In general, the reads can easily understand the source of these missing’s; hence, your suggestion is incorporated in the revised manuscript both in table 1& 2.

3. In table 2, “Status of current pregnancy” could be called “reproductive intention” 

Authors’ response: Amended as recommended

Not convinced that Fig 2 is necessary. 

Authors’ response: based on your recommendation after evaluating its importance figure 2 is removed from the revised manuscript because it does not give any additional benefit for the manuscript as well as for the readers. 

4. Table 3: I find it confusing that the reference group is not listed first in this table – I have not seen results presented in this way before but could be mistaken. I think it would be useful to see the p-value for the multivariate regression models presented in table 3. 

Authors’ response: thank you for your concern. Regarding the first concern; as far as I know, there is no standard to put the reference category at first or last. However, the selection of the reference category is based on the proportion of the outcome of interest (i.e. based on the cross-tabulation proportions; better to consider the variable with low proportion as a reference category). Thus, during the selection of the reference category, we consider those with a low proportion of the outcome variable. The second comment is accepted as it is and the P-value of both the bivariate and multivariable analyses are incorporated after inserting extra columns. 

 Abstract 

5. I do not feel that the first reviewer’s concerns about the abstract have been addressed. There needs to be a stronger rationale for the current study, and the conclusion needs to speak to the results. The current conclusion reads as though a mindfulness intervention for improving mental health was undertaken alongside improving household food security. 

Authors’ response: the abstract, particularly the introduction and conclusion sections, is further revised. In the revised file, the rationale of the study is clearly stated and read as “Prenatal anxiety and depression are the major health problems all over the world. The negative sequela of prenatal comorbid anxiety and depression (CAD) has been suggested to be higher than those of anxiety or depression alone. CAD increases the odds of preterm birth, low birth weight, prolonged labor, operative deliveries, postpartum psychiatric disorders and long term cognitive impairment for the newborn. Despite its significant ill consequences, there is a dearth of studies in low-and middle-income countries; likewise, to the best of our knowledge, no study assessed the prevalence of CAD in Ethiopia. Hence, the purpose of this study was to assess CAD and associated factors among pregnant women in Arba Minch Zuria district, Gamo zone, southern Ethiopia.” Furthermore, the conclusion is reported based on the modifiable risk factors for comorbid anxiety and depression and read as “in general, one in every ten women faced CAD. Nutritional interventions, early identification and treatment of pregnancy-related illness and medical ailments, prenatal psychological screening and interventions are imperative to minimize the risk of CAD in pregnant women.”

 Introduction 

6. I do not feel that the prevalence of mental disorders in LMIC/ African antenatal populations was adequately discussed in the introduction, and noticed that there were African studies of mental health (including Ethiopian) which was not considered/cited. Authors cite Ethiopian studies in Line 66-68 – it would be interesting to know prevalence’s found in these studies for context and comparison. 

I realise that the literature on comorbid depression and anxiety is sparse, but there are some studies which may be useful: 

Sawyer A, Ayers S, Smith H. Pre- and postnatal psychological wellbeing in Africa: A systematic review. Journal of Affective Disorders 2010; 123(1–3): 17-29. 

Baron EC, Hanlon C, Mall S, et al. Maternal mental health in primary care in five low- and middle-income countries: a situational analysis. BMC health services research 2016; 16: 53. 

Redinger S, Pearson RM, Houle B, Norris SA, Rochat TJ. Antenatal depression and anxiety across pregnancy in urban South Africa. Journal of affective disorders 2020; 277: 296-305. 

Biaggi A, Conroy S, Pawlby S, Pariante CM. Identifying the women at risk of antenatal anxiety and depression: A systematic review. Journal of Affective Disorders 2016; 191: 62-77. 

(for medical condition comparison) Sowa NA, Cholera R, Pence BW, Gaynes BN. Perinatal depression in HIV-infected African women: a systematic review. J Clin Psychiatry 2015; 76(10): 1385-96. 

Authors’ response: first of all thank you for your critical review and very useful article suggestion. In the current appearance, the introduction section is almost changed after incorporating the recent accessible published evidence including your recommended articles. Regarding, studies conducted in Ethiopia, the studies merely address antenatal depression and the prevalence of depression is ranged from 11.8% to 31.2% and all the references are appropriately cited.

7. Line 45: Given the wide disparity in prevalences of depression and anxiety stated in the introduction, it would be helpful to have more information about the studies you cite when describing rates of comorbidity – for example, were participants from HIC/LMIC or low/high-risk pregnancies, etc. There is an increasing trend in comorbid anxiety and depression (CAD); Falah et.al. reported that 9.5% of 25,592 participants had self-reported 46 antenatal anxiety and depressive symptoms 

Authors’ response: such misleading statements are critically reviewed and modified accordingly. Regarding the studies included in the meta-analysis conducted by Falah and coworkers most of the studies are from high-income countries; only three studies were included from Africa [1]. Besides, the introduction section is now written chronologically from the global to the local context. 

Lines 50-55 explains the factors associated with “comorbid anxiety and depression” in the literature however only the bulk of cited literature is for either depression or anxiety. Might be useful to describe factors associated with one or the other in Africa/ Ethiopia specifically (as they are slightly different) and then tell the reader what is associated with comorbidity (this literature is sparse, might need to draw from HIC literature). 

 Authors’ response: Because of limited published evidence risk factors associated either with depression or anxiety were included. Almost all risk factors associated with depression/anxiety are also associated with comorbid anxiety and depression. In the revised file articles that assess the risk factors for comorbid anxiety and depression are cited. 

 Methods 

8. When using the PHQ-9 as a measure of probable depression, a cut-off score of 10 is what has been shown in validation studies to represent a depression, including in antenatal populations. I know that a cutoff of >5 has been used in Ethiopia before, and I assume the authors have based their analysis on this however the reasoning given by the authors in the manuscript is confusing and makes it seem like prevalence’s are being under-estimated. I would reconsider this explanation. The same is true for the description of the GAD-7 cutoff and analysis.

Line score of ≥5 was used as a cut-off point for possible antenatal depression [23] due to small/null observations under moderate and moderately severe and severe depression. 

Authors’ response: Primarily we plan to run multinomial logistic regression after categorizing the outcome variable as free of depression, mild, moderate, moderately severe and severe” based on WHO recommendation. However, when we run the frequency the observations were very low under moderate and moderately severe and almost null in the severe category. Then we decided to run a binary logistic regression by recoding the outcome variable to dummy variable as free of depression and with depression. Regarding the cut-of-point used as you mention, previously published studies in developed and some developing countries used a sore cut of point of 10. In our case, the PHQ Amharic version is validated in our countries context and a score of >=5 was considered as the optimal score for increased probability of major depressive disorder and used in Ethiopian studies [2,3]. Similarly, anxiety was categorized as mild, moderate and severe anxiety with cut points of 5, 10, and 15 respectively. But due to the small number of observations under moderate and severe anxiety, it was taken as a dummy variable with a cut-point of ≥ 5 [4]. In the revised, manuscript such inconsistencies and unnecessary statements were carefully addressed and amended.

 Results 

9. Lines 229 -231 Please state what the confounding factors in the model were? 

Authors’ response: as it is reported in Table 3 both the unadjusted (Crude) OR and adjusted OR are reported. Those variables with a p-value of < 0.25 were candidates for multivariable analysis. After adjustment, some variables such as intimate partner violence, social support, and substance abuse were not significantly associated with comorbid anxiety and depression despite they were significantly associated in the bivariate analysis. Also, to assess whether the association is true for the rest of the variables we have to adjust all possible risk factors that have been associated with the outcome of interest. Let us explain it further: for example, household food insecurity is significantly associated with Comorbid anxiety and depression in the bivariate analysis (Crude OR = 5.32, 95% CI (3.13, 9.03) P < 0.001. The risk persists even after adjustment (adjusted OR= 3.51, 95% CI (1.85, 6.64) P < 0.001 for other known risk factors such as medical illness, marital status, a pregnancy danger sign, social support and intimate partner violence and wealth index. Hence, when we assess the association between household food insecurity and CAD the rest of the variables were considered as confounders. This explanation also works for the rest of the variables included in the final model. 

Normal convention for reporting regression results: AOR 95% CI [CI-CI] p-value 

 Authors’ response: The P-value for the unadjusted and adjusted Odds ratio is presented in the final regression model. Adding the P-value in the text makes the sentence lengthy and boring for reading and we linked the text and readers can get the P-value from Table-3.

10. The fact that women in the highest socioeconomic bracket had higher odds of having comorbid depression and anxiety warrants a more detailed explanation/discussion – especially since women reporting food insecurity were at higher risk. It would be interesting to look at/ mention what the unadjusted results were for this variable to see if they were in the same direction, or if there was collinearity with the food insecurity variable in the multivariate regression. 

Authors’ response: we asked this question many times at college and university level symposium presentations. We run the analysis many times and the association persists even after adjustment. The unadjusted OR with 95% CI and P-value for each variable are displayed in Table 3. Furthermore, as it is written under the statistical subsection of the method, multicollinearity was checked using standard error (SE); which is 1.14, 0.68 and 1.27 for household food insecurity, medium and high wealth quintile respectively. All these values are < 2, which confirms there is no collinearity. 

Discussion 

 Lines 260 and 261 – consider referencing. 

Authors’ response: the statement is appropriately cited.

11. Generally I echo the comments of the previous reviewer on the discussion section – that it is fairly written but would benefit from situating results within the LMIC then African then Ethiopian context. 

 Authors’ response: The discussion section, particularly the magnitude part is modified simultaneous with the introduction. In the revised file, additional references were accessed and cited appropriately by following the global to local chronological order. 

12. Nice papers to look at when considering the results of food insecurity: 

Trujillo J, Vieira MC, Lepsch J, et al. A systematic review of the associations between maternal nutritional biomarkers and depression and/or anxiety during pregnancy and postpartum. Journal of Affective Disorders 2018; 232: 185-203. 

Abrahams Z, Lund C, Field S, Honikman S. Factors associated with household food insecurity and depression in pregnant South African women from a low socio-economic setting: a crosssectional study. Social psychiatry and psychiatric epidemiology 2018; 53(4): 363-72.

Authors’ response: Thank you for suggesting these excellent articles; the second paper is significant for our paper and cited in the revised manuscript. The first one is interesting but it is not related to our study.

Reviewer #4: In my opinion, the authors have addressed all the comments and suggestions made by the reviewers and the manuscript could be published

Authors’ response: thank you for your suggestion

Reviewer #5: (No Response)

Authors’ response: I want to say thank you both the reviewers and academic editor for your critical review and suggestion; all the issues raised were carefully addressed. Still if there is any unclear or partially address concern we will amend further.

 References

1. Falah-Hassani K, Shiri R, Dennis C-L (2017) The prevalence of antenatal and postnatal co-morbid anxiety and depression: a meta-analysis. Psychological medicine 47: 2041-2053.

2. Bitew T, Hanlon C, Kebede E, Medhin G, Fekadu A (2016) Antenatal depressive symptoms and maternal health care utilisation: a population-based study of pregnant women in Ethiopia. BMC Pregnancy Childbirth 16: 301.

3. Gelaye B, Williams MA, Lemma S, Deyessa N, Bahretibeb Y, et al. (2013) Validity of the patient health questionnaire-9 for depression screening and diagnosis in East Africa. Psychiatry research 210: 653-661.

4. Zhong Q-Y, Gelaye B, Zaslavsky AM, Fann JR, Rondon MB, et al. (2015) Diagnostic validity of the Generalized Anxiety Disorder-7 (GAD-7) among pregnant women. PloS one 10.

---

## [Editor Report · Decision Letter 2]

25 Feb 2021

Comorbid anxiety and depression: prevalence and associated factors among pregnant women in Arba Minch zuria district, Gamo zone, southern Ethiopia

PONE-D-20-08361R2

Dear Dr. Bante,

We’re pleased to inform you that your manuscript has been judged scientifically suitable for publication and will be formally accepted for publication once it meets all outstanding technical requirements.

Kind regards,

Susan A. Bartels, MD, MPH, FRCPC

Academic Editor

PLOS ONE
---

## [Editor Report · Acceptance letter]

1 Mar 2021

PONE-D-20-08361R2 

Comorbid anxiety and depression: prevalence and associated factors among pregnant women in Arba Minch zuria district, Gamo zone, southern Ethiopia 

Dear Dr. Bante:

I'm pleased to inform you that your manuscript has been deemed suitable for publication in PLOS ONE. Congratulations! Your manuscript is now with our production department. 

Kind regards, 

on behalf of

Dr. Susan A. Bartels 

Academic Editor

PLOS ONE